# INTERPRETABLE COMPACT CATEGORICAL FEATURES ENCODING FOR SUPERVISED LEARNING

## ABSTRACT

In supervised learning, encoding techniques for continuous features are well studied. However, few are specific for categorical features. The categorical encoding approaches that are widely used are one-hot encoding, target encoding and its variant ordered target encoding. In many cases categorical features carry significant, if not a dominant portion of, feature information in a supervised learning problem. Therefore they are key to improve model fit. One hot encoding is known for its curse of dimensionality issue, especially when categorical features are of high cardinality and/or sparse. Such problem not only increases the problem size but may also introduce instability for numerical solvers. Target encoding and its variant ordered target encoding are often used to address such data issues. It is fast and compact. The downside is that target encoding tend to overfit due to the way it is implemented. To our knowledge, the other categorical encoding methods used in machine learning and deep learning algorithms do not preserve interpretability. Our goal is to bridge the gap between dimension reduction, accuracy, feature interpretability, and scalability. In this paper, we introduce a polynomial algorithm called Interpretable Compact Categorical Feature Encoding for Supervised Learning (ICFESL). Under reasonable assumption, our encoding technique ensures no information loss for regression and minimum information loss for classification. At the core, it leverages L2 regularized linear models to efficiently calculate coefficients for one-hot-encoded categorical features and group them together without transforming them. We prove that applying K-means clustering for the grouping problem yields optimal solutions. We test our algorithms on simulations and real world datasets both in regression and classification to validate the assumption and demonstrate the encoding method's performance. The results show that for regressions, ICFESL enabled linear models and xgBoost models often significantly outperform state-of-the-art algorithms such as CatBoost and TabNet in terms of RMSE. The results also show that for classifications, ICFESL has comparable performance and outperforms CatBoost measured by AUC when ordered target encoding shows significant overfitting. We demonstrate how interpretability is preserved with example clusters from one of the experiments.

## 1 INTRODUCTION

In many industrial applications, people train regression and classification models to make informed business decisions. Today, there are many different algorithms that are being used by ML practitioners: from simple models such as linear regression and logistic regression; to up-to-date tree based ensemble models such as XGBoost (Chen & Guestrin, 2016), LightGBM (Ke et al., 2017), CatBoost (Prokhorenkova et al., 2018) ; to more recent deep learning models such as TabNet (Arik & Pfister, 2021) to solve tabular learning problems. The priorities when developing and deploying a ML solution often fall into balancing and optimizing the following aspects: the time and space complexity for model development, speed at model inference and last but not the least the ability to digest and explain model outputs quickly. Categorical data plays a significant role in real world applications and therefore there is tremendous interest of how to improve its encoding. In some industries, categorical feature encoding not only has great importance at model fit and inference but also is often critical to downstream overhead cost and operational efficiency. An example is that when a bank turns down a custom for his/her application for a credit card, the bank is obligated to explain the

reason for rejection in broad categories. Another example is that when analyzing and interpreting customer behavior for debt payment amount, how to efficiently group and visualize based on categories. Faster analysis with high accuracy wins business. In the landmark paper (Lundberg & Lee, 2017), the authors proposed SHAP values as a unified way for model interpretability and is widely adopted in the industry. Despite the fact that kernel SHAP can be applied to any ML or deep learning models for interpretability, it has exponential time complexity and therefore is expensive to scale up. For such reason, today when learning tabular data, where inferencing speed and interpretability are prioritized, the prevailing techniques are still one hot encoding and target encoding or its variant (Poslavskaya & Korolev, 2023). We would like to weigh on the pro and cons of these techniques and discuss how our work contribute to this topic. One hot encoding is a simple transformation that uses a binary vector to represent each level of a categorical feature. The problem comes when categorical features have high cardinality. That is where target encoding and order target encoding comes into place. By replacing categories with mean target values target encoding keeps the target information and saves the space. However, this introduces target leaking and causes overfitting. Its variant ordered target encoding is a step to reduce the leaking concern. However, the overfitting issues is still present, especially when there is new data with unknown categories. In LightGBM (Ke et al., 2017), sparse columns are combined based on a heuristic called Exclusive Feature Bundling, which may be considered as a trade-off between target encoding and OHE. ICFESL starts with the OHE of the categorical features, encodes and collapses the data once (not transform data so interpretability is preserved). To resolve the overfitting issue with unknown categories, ICFESL uses two techniques: L2 regularization to reduce the impact of noisy data in the training data; And a weighting schema based on the number of observations to account for sparse / unknown categories. We organize the paper as following: we will first review existing research in the literature, which connects to several areas such as dimension reduction, model interpretability, feature engineering in general for different ML algorithms. We would like to focus on the intersection of these areas, and focus on interpretable categorical features encoding or embedding techniques. Next we set up the problem statement, derive the foundation for ICFESL and propose the algorithm with L2 regularization fine tuning. Then we put together simulation experiments as well as experiments with real world data to discuss the properties and the performance of the algorithm, compared with state-of-the-art gradient boosting and deep learning algorithms in the industry. Finally, we use a simple example to show how information is extracted and preserved from data for interpretability.

## 2 RELATED WORK

Feature encoding or embedding has been an active research area among the machine learning community and many results exist. However, few methods aim at categorical features, or they work well in general feature embedding but does not have a focus on interpretability. Ordered target encoding for CatBoost and entity embedding (Guo & Berkhahn, 2016) for TabNet, are the most recent developments and represent the state-of-the-art for tree-based models and deep neural network models that solve the same problem. Aside from these algorithms, there are established work in the agnostic model interpretability space such as LIME (Ribeiro et al., 2016) and SHAP (Lundberg & Lee, 2017) to understand grey box and black box models. The main idea in these pioneering work when approaching interpretability is to locally approximate the estimated equation (LIME) or sample/perturb model prediction to inference feature sensitivity. These methods leverage linear models or simulation to fit an additive model as surrogate to explain the original model. Other related papers in the literature include unsupervised feature clustering based on different distance metrics and test statistics (Hu et al., 2022) between binary vectors (Pratap et al., 2019), (Bera et al., 2021). One in particular is clustering categorical features based on hamming distance. Hamming distance between two binary vectors $x$ and $y$ is defined as:

$$HD(x,y) = \frac{1}{n} \sum_{i=1}^{n} dist(x_i, y_i), \text{ where}$$

$$dist(x_i, y_i) = \begin{cases} 1 & \text{if } x_i \neq y_i \\ 0 & \text{otherwise} \end{cases}$$

This approach is the closest to our approach as it can also be used to cluster OHE categorical features. We use it to represent the class of distance-based clustering algorithms that are related. There

exist search based clustering algorithms too. In (Carrizosa et al., 2021) the author proposed an algorithm to cluster levels in a single categorical feature with a semi-heuristic search process Greedy Randomized Adaptive Search Procedure (GRASP) to find a best performing schema measure by out-of-sample performance based on a pre-determined resulting number of clusters K. The GRASP procedure is repeated multiple times and the best schema in the candidate pool is returned. According to (Silva, 1996), GRASP solves a NP-complete problem, which means its complexity is exponential to the number of feature levels or number of rows. Another difference is that ICFESL is designed to run on multiple features. We compared ICFESL with GRASP on the mushrooms dataset: ICFESL has 100 percent accuracy on the mushrooms dataset, GRASP has slightly lower accuracy at 99.85 percent. Also, we would like to make the connection that our work is closely related to the research along the model-agnostic interpretability line (Ribeiro et al., 2016) (Lundberg & Lee, 2017). In (Ribeiro et al., 2016), the author proposed a L2 regularized regression framework to approach local model interpretability. In (Lundberg & Lee, 2017), SHapley Additive exPlanations was introduced for global feature importance and model-agnostic prediction attribution for main effects. We choose the same way to tackle a difficult problem from a more approachable angle: leveraging the upsides of linear models to extract feature information for collapsing. We attempt to tackle feature engineering and interpretability at the early model building stages, before a model is actually fit. We believe this approach can significantly shorten the model development time and increase model inference efficiency. Our approach has several advantages: first, our encoding algorithm is automatic from end to end. The algorithm determines the number of resulting level clusters when collapsing a categorical feature and the optimal clustering based on L2 regularization; second, we prove that our solutions to the problem are feasible and optimal for both regression and classification problems; third, our encoding algorithm can run on multiple categorical features at a time; fourth, our algorithm runs in polynomial time and with no loss of information for regression and minimum loss for classification. Finally, we would like to claim that some more complex learning problems can be cast into the basic form we solve. Therefore the algorithm could find applications outside of its categorical feature setting. In the experiments, we run our algorithm on simulated data as well as real world datasets and use the encoded data to fit different models. We argue that the single assumption we made is reasonable in practice and even it is violated to a large degree the encoding method still yields superior model fit results.

## 3 PROPOSED ALGORITHM

Consider the following supervised learning problem with categorical features only: we have a set of observations $(y, X)$ where $y \in \mathcal{R}^n$ and $X = [X_1, X_2, ..., X_m]$ is the set of categorical features that have $[C_1, C_2, ..., C_m]$ levels respectively. We aim to find $\beta \in \mathcal{R}^m$ the coefficient vector that best fits the relationship between X and y.

$$y = \begin{cases} X\beta & \text{, for regression} \\ ln(\frac{1}{1+e^{-X\beta}}) & \text{, for binary classification} \end{cases} \tag{1}$$

Let the starting point of our problem be the OHE transformation of $X$ in equation (1). To focus on what's important, we look at the representation of OHE:.

$$X_o = [X_{1,1}, ..., X_{1,C_1}, X_{2,1}, ..., X_{2,C_2}, ..., X_{m,1}, ..., X_{m,C_m}] \tag{2}$$

and we have

$$\beta_o = [\beta_{1,1}, ..., \beta_{1,C_1}, \beta_{2,1}, ..., \beta_{2,C_2}, ..., \beta_{m,1}, ..., \beta_{m,C_m}] \tag{3}$$

where $C_i$ is the number of levels of $X_i$. And $X_{i,1}, X_{i,2}, ..., X_{i,C_i}$ are n-dimensional binary vectors in $X_i$'s OHE representation. For $X_i$, our goal is to find optimal clusters $g_{i,1}, ..., g_{i,\hat{C}_i}$ for its levels $1, ..., C_i$ and we need to do this for $i = 1, ..., m$. We know that OLS and Logit are considered the most interpretable among the algorithms that can solve (1). The idea is to obtain the clusters using their coefficients and then leverage their simplicity for fast interpretation and scoring. L2 penalties are used when obtaining their coefficients. This has two benefits: first regularization is often used

against overfitting to prevent exploding coefficients, it helps to reduce the impact of data issues and improve numerical stability in the clustering process. Second it serves as a hyper parameter to fine tune the size and the bias of the resulting dataset. To lay down the foundation for our algorithm we start by regression and then move onto classification.

**Lemma 1.** *Assume for any two distinct binary vectors in $X_o$ we have $X_{i,j}^T X_{p,q}$ is ignorable when $i \neq p$. If $\beta_o$ in (3) is the ridge regression estimate of (1) given design matrix $X_o$ as in (2), then we have the following clustered coefficients $\{\hat{\beta}_{i,t}\}$ is also the ridge regression estimate of problem (1), if we apply the same clustering schema to $\{X_{i,j}\}$ for j=1,...,$C_i$. In particular, if we set one value to the coefficients in cluster t, the clustered coefficients should be set to:*

$$\hat{\beta}_{i,t} = \frac{\sum_{j \in g_{i,j}}(k_{i,j} + \lambda)\beta_{i,j}}{\sum_{j \in g_{i,j}}(k_{i,j} + \lambda)}, \quad t=1,...,\hat{C}_i$$

*where $\hat{C}_i < C_i$ and $\{g_{i,1}, ..., g_{i,\hat{C}_i}\}$ is the clustering schema for $\{X_{i,1}, ..., X_{i,C_i}\}$ that satisfies the ridge regression optimal solution condition, whose number of observations are given by $k_{i,j}$ (where $X_{i,j} = 1$).*

*Proof.* From the ridge regression coefficients estimate formula we have: $(X_o^T X_o + \lambda I)\beta_o = X_o^T y$. Plug in (2) and (3) we have:

$$(X_o^T X_o + \lambda I)\beta_o = \begin{bmatrix} X_{1,1}^T X_{1,1} + \lambda & X_{1,1}^T X_{2,1} & \cdots & X_{1,1}^T X_{m,C_m} \\ \vdots & \vdots & \vdots & \vdots \\ X_{1,C_1}^T X_{1,1} & X_{1,C_1}^T X_{2,1} + \lambda & \cdots & X_{1,C_1}^T X_{m,C_m} \\ \vdots & \vdots & \vdots & \vdots \\ X_{m,1}^T X_{1,1} & X_{m,1}^T X_{2,1} & \cdots & X_{m,1}^T X_{m,C_m} \\ \vdots & \vdots & \vdots & \vdots \\ X_{m,C_m}^T X_{1,1} & X_{m,C_m}^T X_{2,1} & \cdots & X_{m,C_m}^T X_{m,C_m} + \lambda \end{bmatrix} \begin{bmatrix} \beta_{1,1} \\ \vdots \\ \beta_{1,C_1} \\ \vdots \\ \beta_{m,1} \\ \vdots \\ \beta_{m,C_m} \end{bmatrix}$$

$$= \begin{bmatrix} X_{1,1}^T X_{1,1} + \lambda & & \\ & \ddots & \\ & & X_{m,C_m}^T X_{m,C_m} + \lambda \end{bmatrix} \begin{bmatrix} \beta_{1,1} \\ \vdots \\ \beta_{m,C_m} \end{bmatrix}$$

$$= \begin{bmatrix} k_{1,1} + \lambda & & & & & \\ & \ddots & & & & \\ & & k_{1,C_1} + \lambda & & & \\ & & & \ddots & & \\ & & & & k_{m,1} + \lambda & & \\ & & & & & \ddots & \\ & & & & & & k_{m,C_m} + \lambda \end{bmatrix} \begin{bmatrix} \beta_{1,1} \\ \vdots \\ \beta_{1,C_1} \\ \vdots \\ \beta_{m,1} \\ \vdots \\ \beta_{m,C_m} \end{bmatrix}$$

$$= \begin{bmatrix} (k_{1,1} + \lambda)\beta_{1,1} \\ \vdots \\ (k_{1,C_1} + \lambda)\beta_{1,C_1} \\ \vdots \\ (k_{m,1} + \lambda)\beta_{m,1} \\ \vdots \\ (k_{m,C_m} + \lambda)\beta_{m,C_m} \end{bmatrix}$$

So under the assumption, the equation now is:

$$(X_o^T X_o + \lambda I)\beta_o = \begin{bmatrix} (k_{1,1} + \lambda)\beta_{1,1} \\ \vdots \\ (k_{1,C_1} + \lambda)\beta_{1,C_1} \\ \vdots \\ (k_{m,1} + \lambda)\beta_{m,1} \\ \vdots \\ (k_{m,C_m} + \lambda)\beta_{m,C_m} \end{bmatrix} = \begin{bmatrix} X_{1,1}^T y \\ \vdots \\ X_{1,C_1} y \\ \vdots \\ X_{m,1} y \\ \vdots \\ X_{m,C_m} y \end{bmatrix} = X_o^T y$$

Without loss of generality, we can examine a simple case where we cluster the binary vectors obtained from the OHE of $X_1$ into two groups: $g_{1,1} = \{1, ..., t\}$ and $g_{1,2} = \{t+1, ..., C_1\}$. We implement this by setting the coefficients within each group equal. Note that the above equation must be also true for the clustered representation. So we have:

$$\begin{bmatrix} \hat{\beta}_{1,1} \sum_{j \in g_{1,1}} (k_{1,j} + \lambda) \\ \hat{\beta}_{1,2} \sum_{j \in g_{1,2}} (k_{1,j} + \lambda) \end{bmatrix} = \begin{bmatrix} \sum_{j \in g_{1,1}} X_{1,j}^T y \\ \sum_{j \in g_{1,2}} X_{1,j}^T y \end{bmatrix} = \begin{bmatrix} \sum_{j \in g_{1,1}} (k_{1,j} + \lambda)\beta_{1,j} \\ \sum_{j \in g_{1,2}} (k_{1,j} + \lambda)\beta_{1,j} \end{bmatrix}$$

It is obvious from here that:

$$\hat{\beta}_{i,t} = \frac{\sum_{j \in g_{i,t}} (k_{i,j} + \lambda)\beta_{i,j}}{\sum_{j \in g_{i,t}} (k_{i,j} + \lambda)}, \text{ where } t = 1, 2, ..., \hat{C}_i \tag{4}$$

$\square$

A couple of comments are in order. Problem (1) is general and many more complex cases, including continuous features being discretized, can be converted into (1). The denominator of (4) sums up to the total number of observations where $X_{i,j} = 1$ and $j \in g_{i,t}$. When data is sparse and of high cardinality the best encoding we know is ordered target encoding. Using ICFESL means the assumption approximately holds. In practice that is rare by default. However, we argue that it is a reasonable prerequisite. We expect at this stage the data has been treated to avoid numerical issues and severe overfitting, which means constant and near constant columns are removed and highly correlated features are either discarded or dealt with through feature engineering. Alternatively, a quick run through the OHE transformation with low variance filter or a hamming distance clustering algorithm can effectively uphold the assumption. Because of this, we do not expect serious violation of the assumption in practice. To fully understand the impact of the assumption we carefully designed a set of simulations in the experiment section. We choose L2 penalty over L1 because our main objective is to have reliable coefficient estimates rather than subsetting features. Another main factor is that L1 often lacks the numerical stability L2 provides. Before further discussion, let us focus on deriving the important results first. Based on lemma 1 above we are ready to show the main result for regression.

**Theorem 1.** *In the same problem setup as in Lemma 1, given the number of clusters and using t as label, applying K-means clustering to $\{\beta_{i,j}\}, j = 1, ..., C_i$, with $k_{i,j} + \lambda$ as the sample weight for $\beta_{i,j}$ yields the optimal clustering schema. Replacing the clustered coefficients with the weighted version of their cluster centroids gives the reduced equivalent ridge regression solution of (1).*

*Proof.* Let us consider grouping $[\beta_{i,1}, ..., \beta_{i,C_i}]$ into $\hat{C}_i$ clusters $g_{i,1}, ..., g_{i,\hat{C}_i}$. The objective of K-means is to find $\hat{C}_i$ cluster centroids $z_{i,t}, t = 1, ..., \hat{C}_i$ such that the total sum of squares of distance within the clusters is minimized.

$$\arg\min_{g_{i,t}} \sum_{t=1}^{\hat{C}_i} \sum_{j \in g_{i,t}} (z_{i,t} - \beta_{i,j})^2 \tag{5}$$

The centroids are then given by:

$$z_{i,t} = \frac{\sum_{j \in g_{i,t}} (k_{i,j} + \lambda)\beta_{i,j}}{\sum_{j \in g_{i,t}} (k_{i,j} + \lambda)} = \hat{\beta}_{i,t}, \quad t = 1, 2, ..., \hat{C}_i$$

If we repeat this process m times to cluster all OHE coefficients $\{\beta_{i,j}\}, i = 1, ..., m$, then base on Lemma 1, we have $\{z_{i,t}\}$ as the reduced form L2 estimate $\{\hat{\beta}_{i,t}\}$, and the optimal clusters are given by $\{g_{i,t}\}$. $\qquad\square$

We next show the results for classification. Similarly, let us start from the OHE transformation $X_o$. The negative log-likelihood function of problem (1) in a ridge logistic regression is given by

$$\mathcal{L}(\beta_o) = -\frac{1}{n}[\sum_{r=1}^{n} y_r ln(p(x_r, \beta_o)) + (1 - y_r)ln(1 - p(x_r, \beta_o))] + \frac{\lambda}{2}\beta_o^T \beta_o \qquad (6)$$

where $r = 1, ..., n$ and $x_r$ is row r in $X_o$ and $p(x_r, \beta_o) = \frac{e^{x_r \beta_o}}{1+e^{x_r \beta_o}}$. Starting now we use subscript r to indicate the rth row of $X_o$ and let $x_{r,i,j}$ be the rth entry of $X_{i,j}$.

**Theorem 2.** *Assume for any two distinct binary vectors in $X_o$ we have $X_{i,j}^T X_{p,q}$ is ignorable when $i \neq p$. If $\beta_o$ in (3) is the optimal solution minimizing (6) given design matrix $X_o$ as in (2), the following clustered coefficients $\{\hat{\beta}_{i,t}\}$ minimizes the prediction sum of square errors when we set the coefficients in a cluster to equal:*

$$\hat{\beta}_{i,t} = \frac{\sum_{j\in g_{i,t}} \sum_{r\in h_{i,j}} (\mathcal{P}_r')^2 \beta_{i,j}}{\sum_{j\in g_{i,t}} \sum_{r\in h_{i,j}} (\mathcal{P}_r')^2}, \quad t=1,..., \hat{C}_i$$

*where $\hat{C}_i < C_i$ and $\{g_{i,1}, ..., g_{i,\hat{C}_i}\}$ is the clustering schema for $\{X_{i,1}, ..., X_{i,C_i}\}$ that minimizes the prediction sum of square errors. $\mathcal{P}_r'$ is the derivative of the logistic function at $x_r\beta_o$. $h_{i,j} = \{r : x_{r,i,j} = 1\}$ is the set of observations where $X_{i,j} = 1$.*

*Proof.* Let $\mathcal{P}_r' = \mathcal{P}'(x)|_{x=x_r\beta_o} = \frac{e^x}{(1+e^x)^2}|_{x=x_r\beta_o}$ and $\delta_{r,i,j} = x_{r,i,j}(\hat{\beta}_{i,j} - \beta_{i,j})$. where $j \in g_{i,t}$. We can define the square error of the two predictions of the r th row of (1) with $\beta_o$ and $\hat{\beta}$ as:

$$\epsilon_r^2 = (\mathcal{P}_r')^2(\sum_{i=1}^{m}\sum_{j=1}^{C_i}\delta_{r,i,j})(\sum_{i=1}^{m}\sum_{j=1}^{C_i}\delta_{r,i,j}) \qquad (7)$$

Note that $\delta_{r,i,j} = 0$ when $x_{r,i,j} = 0$ and $\hat{\beta}_{i,j} - \beta_{i,j}$ when $x_{r,i,j} = 1$. Since the indicator vectors $X_{i,j}$ sum up to 1 at each row r for fixed i, we have $\delta_{r,i,j_1}\delta_{r,i,j_2} = 0$. Further more, breaking down by different combinations of columns we have:

$$\sum_{r=1}^{n} \delta_{r,i,j}\delta_{r,p,q} = \begin{cases} 0, & i = p \text{ and } j \neq q \\ \sum_{r\in\{r:x_{r,i,j}=1\}}(\hat{\beta}_{i,j} - \beta_{i,j})^2, & i = p \text{ and } j = q \\ \sum_{r\in\{r:x_{r,i,j}=1 \text{ and } x_{r,p,q}=1\}}(\hat{\beta}_{i,j} - \beta_{i,j})(\hat{\beta}_{p,q} - \beta_{p,q}), & i \neq p \end{cases} \qquad (8)$$

Since we assumed $X_{i,j}^T X_{p,q}$ is ignorable the last term in the above equation goes away. We have discussed the reasonableness of this assumption after theorem 1. We invite the reader to the experiments section for an empirical validation experience. Let $h_{i,j} = \{r : x_{r,i,j} = 1\}.i = 1, ..., m.$ and $j = 1, ..., C_i$. Plus in (7) into $\epsilon_r^2$, we have:

$$\sum_{r=1}^{n}\epsilon_r^2 = \sum_{i,j}\sum_{r\in h_{i,j}}(\mathcal{P}_r')^2(\hat{\beta}_{i,j} - \beta_{i,j})^2$$

$$= \sum_{i=1}^{m}(\sum_{t=1}^{\hat{C}_i}\sum_{j\in g_{i,t}}\sum_{r\in h_{i,j}}(\mathcal{P}_r')^2(\hat{\beta}_{i,t} - \beta_{i,j})^2)$$

For each $i = 1, ..., m$ for $X_i$, we solve the following K-means clustering problem:

$$\arg\min_{g_{i,t}} \sum_{t=1}^{\hat{C}_i}\sum_{j\in g_{i,j}}\sum_{r\in h_{i,j}}(\mathcal{P}_r')^2(\hat{\beta}_{i,t} - \beta_{i,j})^2 \qquad (9)$$

From the optimal solution we have:

$$\hat{\beta}_{i,t} = \frac{\sum_{j \in g_{i,t}} \sum_{r \in h_{i,j}} (\mathcal{P}'_r)^2 \beta_{i,j}}{\sum_{j \in g_{i,t}} \sum_{r \in h_{i,j}} (\mathcal{P}'_r)^2} \tag{10}$$

$\square$

Theorem 2 servers as the foundation for our algorithm for classification problems. Since there is no analytical form for $\hat{\beta}$, unlike the regression problem, we use the prediction sum of square errors as the way to measure compression information loss. By design, K-means clustering drives that loss to 0.

---

**Algorithm 1** Interpretable Compact Categorical Features Encoding for Supervised Learning:

---

**Require: Input Data and parameters**:
     - Data after OHE $D_{train}, D_{test}$ .
     - Parameters $\lambda[]$ = L2 regularization parameters.
     - Hamming Distance Clustering=False
     - Min Obs= Default value
     - Stop criterion=Auto
**Ensure:** Optimized level clustering for each categorical feature $X_i$: $\{g_{1,t}\}, ..., \{g_{m,t}\}$.
 1: **for** each $\lambda_k$ in $\lambda[]$ **do**
 2:     **if** Hamming Distance Clustering=True **then**
 3:         Preprocess input data with hamming distance clustering
 4:     **end if**
 5:     Run ridge regression/logisticRegression with $\lambda_k$ on $D_{train}$ to obtain coefficients $\{\beta_{i,j}\}$
 6:     **for** $i = 1$ to $m$ **do**
 7:         $\{\tilde{\beta}_{i,j}\}$ is the filtered set of coefficients of $\beta_{i,j}$, based Min Obs.
 8:         calculate weights for $\{\tilde{\beta}_{i,j}\}$ based on theorem 1 or 2
 9:         **while** $1 <= l <= |\tilde{\beta}_{i,j}|$ **do**
10:            $\{g_{i,t}\}$ = KMeans($\tilde{\beta}_{i,j}$, weights, $l$)
11:            save clustering schema($\lambda_k$), $\{g_{i,t}\}$ and StopCriterion($\lambda_k$).
12:            use cluster centroids $\{\hat{\beta}_{i,t}\}$ and $\{\beta_{i,j}\}$ to calculate error(i)
13:            **if** Optimal stopping criterion satisfied **then**
14:               break
15:            **else**
16:               $l = l + 1$
17:            **end if**
18:            **if** Weighted model fit metric decreases two times in a row **then**
19:               break
20:            **end if**
21:         **end while**
22:     **end for**
23:     Save clustering schema schema($\lambda_k$) = $\{g_{1,t}\}, ..., \{g_{m,t}\}$. error = $\sum_{i=1}^{m}$ error(i)
24:     $\hat{D}_{train}$ = ApplyClustering($D_{train}$, schema($\lambda_k$))
25:     Fit model on $\hat{D}_{train}$. Track training metric RMSE/AUC ($\lambda_k$)
26:     $\hat{D}_{test}$ = ApplyClustering($D_{test}$, schema($\lambda_k$))
27:     Run fitted model on $\hat{D}_{test}$ for testing metric RMSE/AUC ($\lambda_k$)
28:     **if** the weighted score of train/test model fit metrics decreases twice in a row **then**
29:         break
30:     **end if**
31: **end for**
32: **return** optimal clustering schema $schema(\lambda*)$ based on best weighted score of model fit metrics.

---

Some comments about the encoding algorithm. The hamming distance clustering preprocessing is optional. We implement it in case there is need to quickly approximate the assumption before

running through the algorithm. The hyper parameter search for optimal $\lambda$ is similar to what we expect for ridge regression. We choose the min-obs parameter to be 10 because 10 is the rule of thumb of the minimum requirement for a reliable coefficient estimate. In the clustering step, the filtered-out coefficients $\{\beta_{i,j}\}/\{\tilde{\beta}_{i,j}\}$ remain unclustered. The algorithm runs in polynomial time complexity for a single $\lambda$. Based on (Zhang et al., 2014) and (Bulso et al., 2019), running regularized OLS or Logit are both polynomial and is one-time in ICFESL for a single $\lambda$ run. Let $C_* = max(C_i)$, with stopping criterion below the worst case K-means clustering on coefficient vector runs on $O(C_*^2)$. So for a single $\lambda$, m categorical features, the time complexity of ICFESL is the bigger of $O(mC_*^2)$) and the time complexity solving a ridge regression. This means that ICFSEL is polynomial and the bottleneck is often the ridge regression / logisticRegression runs on OHE of problem (1). We implemented two stopping criterion: gap static and inertia ratio. Gap statistic (Tibshirani et al., 2001) is a popular way to test the significance of a clustering schema. In our case it tests against a set of random samples drawn from a uniform distribution that has the same support as the coefficients being clustered. In practice, we found the inertia ratio stopping criterion works better. The inner loop K-Means algorithm satisfies optimal condition if the rate of inertia decrease is lower than a threshold or the size of inertia falls below a threshold $\epsilon < 1$. We use the following criterion to define the threshold for inertia:

$$\frac{inertia(i)}{\sum_{t=1}^{\hat{C}_i} |g_{i,t}| * \|z_{i,t}\|^2 + \sum_{j=1}^{C_i} \|\beta_{i,j}\|^2} < \epsilon, \text{ for i = 1,..., m.} \tag{11}$$

The following equation shows how it works:

$$inertia(i) = \sum_{t=1}^{\hat{C}_i} \sum_{j \in g_{i,t}} \|z_{i,j} - \beta_{i,j}\|^2 <= \sum_{t=1}^{\hat{C}_i} \sum_{j \in g_{i,t}} \|z_{i,j}\|^2 + \sum_{t=1}^{\hat{C}_i} \sum_{j \in g_{i,t}} \|\beta_{i,j}\|^2$$

$$= \sum_{t=1}^{\hat{C}_i} |g_{i,t}| * \|z_{i,t}\|^2 + \sum_{j=1}^{C_i} \|\beta_{i,j}\|^2 \tag{12}$$

## 4 EXPERIMENTS

In this section, we carry out simulations to examine the performance of the encoding algorithm against widely used algorithms or encoding methods paired with regression / xgboost. We want to show that the "orthogonal" assumption we made in the main results is reasonable and indeed ignorable. We also put together experiments in the appendix with 5 real world datasets to show its efficiency in practice. The experiment environment is MacBook Air M3 with 16 GB memory. We simulate (1) with two categorical features that each has 1000 levels. The testing parameters are the correlation $\rho$ between the two features. We also simulate noise in the data: regression is the standard deviation $\sigma$ of the white noise and classification is the rate of target flip $\phi$. To test the fitted model on unknown categories we intentionally set aside part of $X_1$ from training dataset and stack them into testing data so that it is unknown at the time of training.

Table 1: Simulation Parameters

| | Formula | $\rho$ | $\sigma$ | $\phi$ |
|---|---|---|---|---|
| Regression 1 | $5 + 10X_1 - 10X_2$ | 0.2 | 50 | . |
| Regression 2 | $5 + 10X_1 - 10X_2$ | 0.5 | 50 | . |
| Regression 3 | $5 + 10X_1 - 10X_2$ | 0.2 | 200 | . |
| Classification 1 | $1 + X_1 - X_2$ | 0.2 | . | 0.03 |
| Classification 2 | $1 + X_1 - X_2$ | 0.5 | . | 0.03 |
| Classification 3 | $1 + X_1 - X_2$ | 0.2 | . | 0.15 |

Table 2: Simulation Experiments Model Fit Metrics Comparison

| RMSE/AUC | Regression 1 | Regression 2 | Regression 3 | Classification 1 | Classification 2 | Classification 3 |
|---|---|---|---|---|---|---|

| | | | | | | | |
|---|---|---|---|---|---|---|---|
| Feature number | CatBoost | 2 | 2 | 2 | 2 | 2 | 2 |
| | TabNet | 2 | 2 | 2 | 2 | 2 | 2 |
| | OHE | 1998 | 1998 | 1998 | 1998 | 1998 | 1998 |
| | Target | 2 | 2 | 2 | 2 | 2 | 2 |
| | ICFESL | 42 | 40 | 42 | 15 | 14 | 16 |
| | Hamming | 861 | 870 | 861 | 884 | 869 | 884 |
| Algorithm Train | CatBoost | 1076 | 1334 | 1087 | 0.96 | 0.93 | 0.85 |
| | TabNet | 560 | 406 | 352 | 0.96 | 0.96 | 0.85 |
| Algorithm Test | CatBoost | 1136 | 1452 | 1144 | 0.94 | 0.89 | 0.80 |
| | TabNet | 574 | 417 | 357 | 0.94 | 0.94 | 0.81 |
| Logit Train | Target | 873 | 1122 | 893 | 0.96 | 0.93 | 0.85 |
| | ICFESL | 214 | 227 | 287 | 0.95 | 0.93 | 0.84 |
| | Hamming | 2748 | 2399 | 2755 | 0.81 | 0.76 | 0.73 |
| Logit Test | Target | 958 | 1232 | 978 | 0.94 | 0.89 | 0.80 |
| | ICFESL | 216 | 227 | 299 | 0.94 | 0.92 | 0.80 |
| | Hamming | 2940 | 2560 | 2946 | 0.77 | 0.70 | 0.69 |
| xgboost Train | Target | 691 | 898 | 727 | 0.97 | 0.95 | 0.88 |
| | ICFESL | 297 | 284 | 354 | 0.95 | 0.93 | 0.84 |
| | Hamming | 3173 | 2661 | 3179 | 0.74 | 0.70 | 0.69 |
| xgboost Test | Target | 815 | 1071 | 861 | 0.94 | 0.90 | 0.80 |
| | ICFESL | 302 | 284 | 368 | 0.94 | 0.92 | 0.80 |
| | Hamming | 3297 | 2757 | 3301 | 0.70 | 0.65 | 0.64 |

Table 1 shows the settings of the experiments and table 2 shows the performance of ICFESL enabled model fit compared with widely used models. First we look at regression: ICFESL significantly outperforms CatBoost in all cases and TabNet in most cases. In regression 2, the correlation is 0.5, which is considered very high if one cares about interpretability. However, based on simulation it does not seem to impact ICFESL performance. In regression 3 we add a lot of noise to the target. ICFESL is on par with TabNet, and outperforms the other algorithms. Feature dimension is compressed from 1998 to 40 by ICFESL with a quick search through hyper parameters. And the prediction sum of square errors is at $e - 13$ level for the 3 cases, as we expect it to be close to zero. Next we look at classification: ICFESL compresses the data more in this case from 1998 to around 15 in feature dimension and it enables comparable performance to CatBoost and TabNet. Correlation again does not impact ICFESL. In fact when correlation is high ICFESL performs better in testing than CatBoost. We think the reason is that ordered target encoding is more likely to overfit in such scenario. The average prediction sum of square errors for classification is at $e - 5$ level. When K-means converges, ICFESL minimize the information loss. We found when there is no numerical issue in the ridge regression process and $\lambda$ is appropriate, the information loss is usually close to zero. We also test ICFESL on 5 real world datasets. One major difference for the real world datasets is that we need to preprocess but we cannot easily set up some categories to be unknown in training but in testing without biasing the experiments, like we can in simulation. We think that is one reason the regression performance of ICFESL is not as sharp as in simulations. We refer to interested readers to the Appendix for more detail. There we also include run time comparison and an example to further illustrate how feature information is extracted and preserved by clustering categorical feature levels. As we would expect, the time complexity of ICFESL is about the same as ridge regression.

## 5 CONCLUSION

Categorical features play a critical role in tabular learning problems but few encoding methods exist specifically for categorical features. The goal of our research is to provide a scalable method that can be integrated into model building for dimension reduction and feature interpretability while retaining accuracy. We propose a polynomial algorithm ICFESL based on theory and include practical considerations. We use simulation to show that the assumption we made is reasonable and does not impact performance. Simulation shows performance is stable and consistent even when high correlation and noise is present. We also test it on real-world datasets with data preprocessing to validate its usage in practice. We conclude that ICFESL can provide significantly better performance in regression problems and is comparable to CatBoost and TabNet in classifications in terms of model fitting metrics, with interpretability largely retained. We hope ICFESL can find applications in early model building stages for different machine learning algorithms, as well as for data compression and visualization.

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

## 6 Appendix

Below shows the results for the 5 experiments using the real world datasets described in the experiments section. There does not seem to be a standard list of such datasets we can use for our research. So we first choose datasets that are frequently cited in the research community. Then we also include datasets that have high cardinal categorical features and are accissible to the broader research community. Table 3 below is an overview of the real world datasets we used: Mushrooms from UCI machine learning repository (UCI Machine Learning Repository); Fraud UW is a benchmark dataset published in proceedings of NeurIPS 2022 (Jesus et al., 2022). Fraud transac (Kaggle, Financial Transactions), US AQI (Kaggle, Air Quality) and Flight Price Prediction (Kaggle, Flight Price) are from Kaggle.com, published by industrial research community. For example, in the Fraud Transac dataset, merchant city along has 12492 levels; in the US AQI dataset, feature local site name has 1033 unique values. Table 3 gives us the summary of the size of the experiment datasets. When using a real-world dataset, preprocessing raw datasets is required, which is part of what we assume. Except for Mushrooms, whose features are all categorical, we preprocess the datasets with segmentation and sampling to ensure the quality and fairness of comparison. Since the algorithm is to encode categorical features we only include categorical features and their OHE in our experiments. In the US AQI dataset, we created two categorical feature month and hour based on date time. And for mushrooms and US AQI, we removed features that have almost zero variance. We include the source code for all experiments with our submission so the process can be reproduced and verified.

Table 3: Dataset Summary

|  | Mushrooms | Fraud UW | Fraud Transac | US AQI | Flight Price Predict |
|---|---|---|---|---|---|
| Type | Classification | Classification | Classification | Regression | Regression |
| Rows | 5686 | 16107 | 55856 | 70000 | 49000 |
| Columns | 21 | 33 | 38 | 29 | 12 |
| Used Cat Features | 21 | 17 | 10 | 8 | 8 |

Table 4 and 5 shows how the problem sizes change based on different encoding techniques and the corresponding metrics for regression and classification, respectively. As we can see, OHE inflates problem size the most. Target encoding works well in many cases and it is the most efficient encoding method. The results show that ICFESL performance is consistent with simulations: ICFESL enabled xgboost performance is stronger than CatBoost and TabNet in several cases and in the rest of the cases it is on par with the other models.

Table 4: Classification Problem Metrics

| Classification | | Mushrooms | Fraud UW | Fraud Transac |
|---|---|---|---|---|
|  | CatBoost | 21 | 17 | 10 |
|  | TabNet | 21 | 17 | 10 |
| Feature number | OHE | 49 | 797 | 935 |
|  | Target | 21 | 17 | 10 |

| | | | | |
|---|---|---|---|---|
| | ICFESL | 34 | 514 | 32 |
| | Hamming | 48 | 345 | 234 |
| Algorithm Train AUC | CatBoost | 1 | 0.87 | 0.99 |
| | TabNet | 1 | 0.84 | 1 |
| Algorithm Test AUC | CatBoost | 1 | 0.85 | 0.99 |
| | TabNet | 1 | 0.82 | 0.99 |
| Logit Train AUC | OHE | 1 | 0.88 | 0.94 |
| | Target | 0.99 | 0.84 | 0.99 |
| | ICFESL | 1 | 0.87 | 0.99 |
| | Hamming | 1 | 0.87 | 0.99 |
| Logit Test AUC | OHE | 1 | 0.79 | 0.94 |
| | Target | 0.99 | 0.85 | 0.98 |
| | ICFESL | 1 | 0.80 | 0.98 |
| | Hamming | 1 | 0.83 | 0.98 |
| xgboost Train AUC | OHE | 1 | 0.92 | 0.99 |
| | Target | 1 | 0.98 | 1 |
| | ICFESL | 1 | 0.91 | 0.99 |
| | Hamming | 1 | 0.92 | 0.98 |
| xgboost Test AUC | OHE | 1 | 0.83 | 0.99 |
| | Target | 1 | 0.83 | 0.98 |
| | ICFESL | 1 | 0.83 | 0.98 |
| | Hamming | 1 | 0.83 | 0.99 |

Table 5: Regression Problem Metrics

| Regression | | US Air Quality Index | Flight Price Predict |
|---|---|---|---|
| Feature number | CatBoost | 8 | 8 |
| | TabNet | 8 | 8 |
| | OHE | 736 | 76 |
| | Target | 8 | 8 |
| | ICFESL | 147 | 35 |
| | Hamming | 563 | 76 |
| Algorithm Train RMSE | CatBoost | 17.1 | 4515 |
| | TabNet | 17.4 | 4461 |
| Algorithm Test RMSE | CatBoost | 17.6 | 4598 |
| | TabNet | 18 | 4479 |
| OLS Train RMSE | OHE | 18.2 | 6669 |
| | Target | 17.6 | 7093 |
| | ICFESL | 17.5 | 6671 |
| | Hamming | 17.5 | 6669 |
| OLS Test RMSE | OHE | 18.6 | 6624 |
| | Target | 18.1 | 7036 |
| | ICFESL | 18 | 6627 |
| | Hamming | 5.7e+10 | 6624 |
| xgboost Train RMSE | OHE | 16.6 | 4082 |
| | Target | 16.2 | 3884 |
| | ICFESL | 16.6 | 4126 |
| | Hamming | 16.6 | 4082 |
| xgboost Test RMSE | OHE | 17.4 | 4431 |
| | Target | 17.2 | 4211 |
| | ICFESL | 17.4 | 4419 |
| | Hamming | 17.4 | 4432 |

Table 6: Estimation Run Time

| Problem | | Mushrooms | Fraud UW | Fraud Transac | US Air Quality Index | Flight Price Predict |
|---|---|---|---|---|---|---|
| Algorithm Training Time | CatBoost | 0.36 | 0.33 | 0.69 | 0.59 | 0.5 |
| | TabNet | 3.7 | 14.8 | 29.8 | 25.9 | 34.5 |
| OLS Training Time | OHE | 0.31 | 21.2 | 20.4 | 7.63 | 0.22 |
| | Target | 0.03 | 0.06 | 0.11 | 0.01 | 0.02 |
| | ICFESL | 0.11 | 9 | 0.63 | 0.57 | 0.06 |
| | Hamming | 0.2 | 5.5 | 13.1 | 2.8 | 0.14 |
| xgboost Training Time | OHE | 0.05 | 0.53 | 1.4 | 2.4 | 0.3 |
| | Target | 0.0002 | 0.23 | 0.23 | 0.26 | 0.17 |
| | ICFESL | 0.13 | 0.91 | 0.32 | 16.6 | 0.3 |
| | Hamming | 0.14 | 0.3 | 0.49 | 1.1 | 0.33 |

At last we show a simple example how ICFESL extracts and preserves interpretability. In the US air quality index experiment, the index is measured and calculated at different location and time with

different method nation wide. Although we are not expert to air quality we expect that air quality is clustered by geographical locations. This is true when we examine the ICFESL clustering scheme from the experiment. For instance Connecticut, Massachusetts, New Hampshire and Rhode Island are in one group. And in another cluster there are Nebraska, South Dakota and Wisconsin.

