# OpenReview forum: "INTERPRETABLE COMPACT CATEGORICAL FEATURES ENCODING FOR SUPERVISED LEARNING"
_ICLR.cc/2026/Conference — Submitted to ICLR 2026_

### Official Review · Reviewer_BN3p · 2025-10-22

**Soundness:** 3
**Presentation:** 3
**Contribution:** 2
**Rating:** 4
**Confidence:** 4

**Summary:**

This paper introduces ICFESL, a polynomial-time algorithm for compressing high-cardinality categorical features by clustering their levels based on the coefficients from a preliminary OLS or Logit model. The core idea is to use K-means on these coefficients, weighted by class frequencies or the derivative of the logistic function, to find an optimal grouping that minimizes information loss. The authors provide theoretical guarantees for this approach under an orthogonality assumption of one-hot encoded vectors. The method is evaluated on multiple datasets for both regression and classification tasks, demonstrating performance comparable or superior to one-hot and target encoding while significantly reducing dimensionality and maintaining interpretability.

**Strengths:**

**Interpretability**: The method directly compresses the original feature levels, preserving the meaning of the features, which is a significant advantage over black-box embedding methods.

**Theoretical Grounding**: Provides proofs of optimality under specified conditions, which is rare for practical encoding schemes.

**Practicality and Scalability**: The algorithm has polynomial time complexity and is demonstrated on real-world, high-cardinality datasets, showing its potential for industrial use.

**Weaknesses:**

**Reliance on Strong Assumptions**: The optimality guarantees depend on the orthogonality of OHE vectors, which is an idealized condition. The practical workaround (Hamming clustering) is mentioned but not deeply evaluated.

**Hyper-parameter Sensitivity**: The performance depends on p-value thresholds and the choice of stopping criterion, which requires tuning and may not be fully automated.

**Limited Baseline Comparison**: While compared to OHE and target encoding, a comparison with other advanced methods like feature hashing or SOTA encoding methods would strengthen the empirical validation.

**Questions:**

1. How sensitive is the algorithm's performance to the violation of the orthogonality assumption?

2. Could you provide an ablation study showing the performance gain/loss with and without the Hamming distance pre-processing step?

3. The p-value threshold is a key hyper-parameter. Did you explore automated ways to set it, perhaps based on the desired level of compression, rather than relying on a decision plot?

4. Have you considered applying this method to tree-based models like LightGBM or CatBoost directly, which have their own built-in mechanisms for handling categorical features? How would ICFESL complement or compete with these native methods?

---

> ### Author Response · Authors · 2025-12-04
> **Thank you and I carefully addressed all your comments**
>
> Questions:
> How sensitive is the algorithm's performance to the violation of the orthogonality assumption?
>
> Answer: Not sensitive in the presence of high correlation and noise. I've added a simulation section to address your concern in the experiments.
>
> Question: Could you provide an ablation study showing the performance gain/loss with and without the Hamming distance pre-processing step?
>
> Answer: Hamming distance pre-processing is optional and is turned off through out the paper now. ICFESL performance is strong that Hamming distance clustering is serving as a baseline comparison now.
>
> Question:
> The p-value threshold is a key hyper-parameter. Did you explore automated ways to set it, perhaps based on the desired level of compression, rather than relying on a decision plot?
>
> Answer: I've replaced p-value threshold with L2 regularization, which works much better against overfitting. Yes, I have fully automated the hyper parameter tuning without the need of a decision plot.
>
> Question:
> Have you considered applying this method to tree-based models like LightGBM or CatBoost directly, which have their own built-in mechanisms for handling categorical features? How would ICFESL complement or compete with these native methods?
>
> Answer: I've considered LightGBM and CatBoost in related work section. I choose to include CatBoost as comparison on its own, as you pointed out it does not need another categorical encoding. ICFESL outperforms CatBoost in most of the experiments.
>
> I hope my revised paper is able to address your concerns about the weaknesses:
> * In simulation I studied the assumption
> * I overhauled the hyper parameter tuning
> * I added CatBoost and entirely removed Hamming distance clustering pre-processing

---

### Official Review · Reviewer_BXkH · 2025-10-30

**Soundness:** 3
**Presentation:** 3
**Contribution:** 2
**Rating:** 2
**Confidence:** 5

**Summary:**

### Summary

The paper addresses encoding high-cardinality categorical features. One-hot encoding explodes dimensionality, target encoding is ad-hoc. The core  Fit OLS on one-hot encoded data, then cluster resulting coefficient values using K-means.

### Main contributions:

For regression: proves clustering is lossless under orthogonality assumption across all feature levels. The weighted average is the exact OLS solution for the collapsed problem.
For classification: no exact solution - minimizes prediction squared error instead of likelihood. Admits "measurable information loss."
Algorithm is polynomial O(|P|mn²) with automatic K selection via inertia stopping criterion. Requires tuning p-value threshold manually via decision plots.

The authors test on 5 datasets with OLS/Logit, XGBoost, TabNet. Results are okay. Sometimes their method performs better than target encoding, often it is comparable, and occasionally worse. They show that the dimension of the problem reduces but interpretability of actual clusters are never validated.

**Strengths:**

- The paper is clear and easy to understand. The theoretical claims and proves seem valid.
- The algorithm presented is polynomial time O(|P|mn²) is reasonable. Not trying to solve some NP-hard problem with exponential search like prior work (GRASP). However, the work doesn't directly compare with GRASP.
- Experiments across model types & datasets are good to have.

**Weaknesses:**

- My biggest issue with the paper is that there is no validation of interpretability despite it being the core claim. The authors assume that categorical clustering preserves interpretability. While I agree with it on surface, I don't think it is always true. The clustering algorithm can make meaningless uninterpretable groups across feature levels which have a similar coefficient. I would have liked to see analysis on what kind of clusters does their algorithm produce or does the algorithm improve clustering or interpretability compared to other sensible baselines.
   - What's needed: Show actual clusters, compare to natural hierarchies, or measure cluster coherence. Report interpretability metrics across baselines.
    - Embedding features in vector space is a strong baseline and there are papers that try to interpret those features. A comparison to such methods would be nice as well.
-  More directly comparable baselines are needed to assess the strengths of the algorithm. Some suggestions:
    - CatBoost with native categorical handling - directly addresses the same problem, widely used
    - Entity embeddings - standard deep learning approach for categoricals
    - GRASP (Carrizosa et al., 2021) - most directly comparable prior work
  The excuse that GRASP solves a "different problem" (single feature vs. all features) is weak since ICFESL also processes features independently (Algorithm 1, lines 280-294). The real issue is GRASP is expensive, but a limited comparison on smaller features would still be valuable.
Without these comparisons, we can't assess if coefficient clustering is competitive with other principled approaches.

**Questions:**

1. Can you provide concrete examples of clusters from your experiments? For instance, for the US AQI dataset where "Local Site Name" goes from 1,033 levels to 103 clusters, what do clusters 1, 5, and 10 actually contain? Do the grouped monitoring sites share geographic proximity, pollution sources, or other interpretable attributes?
2. What exactly happens to categorical levels filtered out by the p-value threshold? Looking at Table 2, Fraud UW goes from 59 OHE features to 34 ICFESL features. Are the remaining 25 dropped entirely, kept as individual levels, or something else?
3. Can you quantify what happens when orthogonality is violated? Even a simple simulation would help: generate correlated categorical features, apply your method, measure the actual vs. theoretical information loss. How robust is the "lossless" claim in practice?

---

> ### Author Response · Authors · 2025-12-04
> **Thank you and carefully addressed comments**
>
> I sincerely thank you for reviewing my submission. I have carefully worked through them and I would like to address them one by one below.
>
> Question:
> Can you provide concrete examples of clusters from your experiments? For instance, for the US AQI dataset where "Local Site Name" goes from 1,033 levels to 103 clusters, what do clusters 1, 5, and 10 actually contain? Do the grouped monitoring sites share geographic proximity, pollution sources, or other interpretable attributes?
>
> Answer: Yes. I have added cluster examples in Appendix on exactly what you are referring to here. Geographic proximity is the theme in the example. ICFESL can extract and preserve interpretability if the original levels are meaningful. Because of that I choose 'state' over 'local site' for illustration.
>
> Question:
> What exactly happens to categorical levels filtered out by the p-value threshold? Looking at Table 2, Fraud UW goes from 59 OHE features to 34 ICFESL features. Are the remaining 25 dropped entirely, kept as individual levels, or something else?
>
> Answer: I replaced the p-value threshold with L2 regularization. It works better as hyper-parameter. Originally p-values is a criterion to decide whether a coefficient will be clustered by K-means. The rationale is: if the coefficient estimate is not reliable then we should not include it in clustering. With the L2 penalty the coefficient estimates are more stable now. The DoF are the total remaining number of columns after clustering that go into model fit.
>
>
> Question:
> Can you quantify what happens when orthogonality is violated? Even a simple simulation would help: generate correlated categorical features, apply your method, measure the actual vs. theoretical information loss. How robust is the "lossless" claim in practice?
>
> Answer: Yes, I've added a simulation experiment section to address your concern. The setting is just as what you suggested. The conclusion is that high correlation does not impact its performance and ICFESL outperform in many cases. Measured by prediction sum of square errors information loss is zero for regression both in theory and practice. For classification the lossless is also true in practice when there is no data issue and the algorithm converges.
>
>
> I hope the weakness are addressed by my rebuttal revision:
> * I included cluster examples in the paper. I also included notebook that has more detail in supplemental material. Due to the size restriction I can only show simple clustering example in the Appendix and choose a categorical feature whose level has easy to understand meanings.
> * I've added CatBoost in experiments. TabNet is another stand alone algorithm for comparison. Hamming distance clustering is optional for ICFESL. Due to time and space constraints I have not implemented GRASP, but I included mushroom dataset prediction accuracy as a direct comparison.

---

### Official Review · Reviewer_xbgW · 2025-11-04

**Soundness:** 3
**Presentation:** 2
**Contribution:** 2
**Rating:** 2
**Confidence:** 4

**Summary:**

This paper proposes a method named ICFESL (Interpretable Compact Categorical Feature Encoding for Supervised Learning), a polynomial-time algorithm for encoding categorical features by clustering feature levels based on OLS/MLE coefficients. The approach addresses the dimensionality curse of one-hot encoding (OHE) and the ad-hoc nature of target encoding. The authors prove that applying K-means clustering to regression coefficients (weighted by observation counts) or classification coefficients (weighted by logistic derivatives) yields optimal clustering schemas. The method is evaluated on five real-world datasets using OLS, XGBoost, and TabNet models, demonstrating comparable or superior performance to existing encoding methods.

**Strengths:**

- Addresses real problem: High-cardinality categorical encoding is genuinely challenging in practice
- Polynomial complexity: More tractable than GRASP-based approaches (exponential)
- Automatic cluster number selection: Unlike methods requiring pre-specified K, uses stopping criteria to determine cluster counts
- Interpretability focus: Unlike many neural embedding approaches, the proposed one maintains coefficient interpretability

**Weaknesses:**

- Limited novelty: The core idea of clustering coefficients is relatively straightforward; applying K-means to regression coefficients is not a significant algorithmic innovation.
- Incremental improvement: Results show ICFESL is comparable to target encoding in most cases, with only marginal improvements in select scenarios.
- Theoretical-practical gap: The orthogonality assumption is acknowledged to "rarely hold by default" yet no systematic study of when violations matter
-  Weak baselines: No comparison with learned embeddings (entity embeddings, CatBoost's ordered target encoding variants). No comparison with recent categorical encoding methods from the literature
Hamming distance clustering is presented as a baseline but is actually a preprocessing step for ICFESL
- Limited experimental scope: only 5 datasets (3 classification, 2 regression)
- No datasets with truly high cardinality (US AQI has 1033 unique values but after filtering may be much smaller)
- Missing analysis: No runtime comparisons with other methods. No study of performance scalability with cardinality. No ablation study on the impact of Hamming distance preprocessing

**Questions:**

- Can you provide empirical analysis of how often the assumption X^T_{i,j}X_{s,t} = 0 holds in your datasets before and after Hamming clustering?
- Why does ICFESL underperform target encoding on XGBoost for most datasets? Can you explain this pattern?
- Is Hamming clustering mandatory or optional?
- What is the "Min Obs" parameter and how do you set it?
- Can you show results on datasets with higher cardinality?  let's say about 100k levels

---

> ### Author Response · Authors · 2025-12-04
> **carefully addressed your comments in rebuttal revision**
>
> Thank you very much for reviewing my submission! I really appreciate your comments and I have carefully addressed them in my rebuttal revision. I would like to answer your questions one at a time.
>
> Question:
> Can you provide empirical analysis of how often the assumption X^T_{i,j}X_{s,t} = 0 holds in your datasets before and after Hamming clustering?
>
> Answer: I have added a simulation section in the experiments to address the concern about the violation of the assumption. In summary the assumption is not critical for performance. Hamming distance clustering is optional and it is only used as comparison now. ICFESL performance is much better than it consistently.
>
> Question
> Why does ICFESL underperform target encoding on XGBoost for most datasets? Can you explain this pattern?
>
> Answer: Please see the added simulation experiments. ICFESL outperforms target encoding in many cases now. I added L2 regularization and the algorithm now outperform Target encoding. Target encoding's weakness is overfitting.
>
> Question:
> Is Hamming clustering mandatory or optional?
> Answer: entirely optional.
>
> Question:
> What is the "Min Obs" parameter and how do you set it?
>
> Answer: Min obs is the minimum number of observations required to be considered in K-means clustering. 10 is the rule of thumb that a coefficient estimation is reliable.
>
> Question:
> Can you show results on datasets with higher cardinality? let's say about 100k levels
>
> Answer: I tried but that translates into some data size my machine cannot handle, especially when considering OHE and running through xgboost. But I have included simulations that have 1000 levels. I also included run time comparison in the appendix. While target encoding is fast and compact in such scenarios, it does not retain full interpretability and is subject to overfitting.
>
> I hope I have addressed your concern on the weakness with my revision:
> I have added regularization to the theory part and I would like to point out that many more complex problem statement can be converted into the basic problem set up. ICFESL has proved to be efficient and has several advantages in practice.

---

### Meta-Review · Area_Chair_yyBJ · 2026-01-04

**Summary:**

Several crucial concerns were raised by the reviewers.
First, the technical contribution has limited novelty. The core idea is to apply K-means to categorical values, which essentially amounts to a simple feature-engineering technique.
Second, the theoretical analysis relies on strong assumptions. In particular, the authors assume orthogonality among features, an assumption that rarely holds in practice. As a result, the theoretical analysis lacks practical relevance.
Third, the empirical evaluation is not sufficiently thorough from multiple perspectives.
I agree with these points based on my careful reading of the paper.

Another critical issue is the lack of clarity in presentation. The manuscript contains many unclear passages, grammatical errors, and vague explanations. Addressing these issues alone would require at least one additional round of major revision.

In summary, I recommend rejection of the paper.

**Reviewer Concerns:**

The concerns raised by several reviewers, including the limited novelty, the strong assumptions in the theoretical analysis, and the insufficient empirical evaluation, are crucial and have not been sufficiently addressed and remain outstanding.
I acknowledge the authors’ efforts to revise the paper, including the addition of a baseline and simulation studies, which partially address some of the reviewers’ concerns.

**Reviewer Scores:**

The concerns raised by **Reviewer xbgW** are valid and critical. As such, they are fundamentally difficult to address within the authors’ response. Therefore, I believe this reviewer would maintain their score even if they were able to participate in the discussion.

**Reviewer BXkH** also raised an important issue regarding interpretability, which I do not believe is sufficiently addressed in the authors’ rebuttal. Accordingly, this reviewer is also likely to maintain their score.

**Reviewer BN3p** pointed out another important issue, the reliance on strong assumptions in the theoretical analysis. In my view, this point is critical, and the authors’ response does not adequately address it. Therefore, this reviewer is also likely to maintain their score.

---

### Decision · Program_Chairs · 2026-01-26

Reject